METHODS AND RESOURCES

# Increasing plant group productivity through latent genetic variation for cooperation

**Samuel E. Wuest**[1,2,3]*, **Nuno D. Pires**[1], **Shan Luo**[4], **Francois Vasseur**[5], **Julie Messier**[6], **Ueli Grossniklaus**[1], **Pascal A. Niklaus**[2]

**1** Department of Plant and Microbial Biology & Zurich-Basel Plant Science Center, University of Zurich, Zurich, Switzerland, **2** Department of Evolutionary Biology and Environmental Studies & Zurich-Basel Plant Science Center, University of Zurich, Zurich, Switzerland, **3** Group Breeding Research, Division Plant Breeding, Agroscope, Wädenswil, Switzerland, **4** Lancaster Environment Centre, Lancaster University, Lancaster, United Kingdom, **5** CEFE, Univ Montpellier, CNRS, EPHE, IRD, Montpellier, France, **6** Department of Biology, University of Waterloo, Waterloo, Canada

* samuel.wuest@agroscope.admin.ch

**Data Availability Statement:** The datasets described and a basic analysis script are available through the Zenodo data repository (DOI:10.5281/ zenodo.6983283).

## Abstract

Historic yield advances in the major crops have, to a large extent, been achieved by selection for improved productivity of groups of plant individuals such as high-density stands. Research suggests that such improved group productivity depends on "cooperative" traits (e.g., erect leaves, short stems) that—while beneficial to the group—decrease individual fitness under competition. This poses a problem for some traditional breeding approaches, especially when selection occurs at the level of individuals, because "selfish" traits will be selected for and reduce yield in high-density monocultures. One approach, therefore, has been to select individuals based on ideotypes with traits expected to promote group productivity. However, this approach is limited to architectural and physiological traits whose effects on growth and competition are relatively easy to anticipate. Here, we developed a general and simple method for the discovery of alleles promoting cooperation in plant stands. Our method is based on the game-theoretical premise that alleles increasing cooperation benefit the monoculture group but are disadvantageous to the individual when facing noncooperative neighbors. Testing the approach using the model plant *Arabidopsis thaliana*, we found a major effect locus where the rarer allele was associated with increased cooperation and productivity in high-density stands. The allele likely affects a pleiotropic gene, since we find that it is also associated with reduced root competition but higher resistance against disease. Thus, even though cooperation is considered evolutionarily unstable except under special circumstances, conflicting selective forces acting on a pleiotropic gene might maintain latent genetic variation for cooperation in nature. Such variation, once identified in a crop, could rapidly be leveraged in modern breeding programs and provide efficient routes to increase yields.

## Introduction

Crop breeding is currently undergoing fundamental transformations. Speed breeding and genomic prediction can shorten generation times and increase effective population sizes,

**Funding:** This work was supported by the University of Zurich, Agroscope and an Ambizione Fellowship (PZ00P3_148223) of the Swiss National Science Foundation (to SEW), and an Advanced Grant of the European Research Council (to UG). FV acknowledges funding from the French Agency for Research (ANR grant ANR-17-CE02-0018-01, 'AraBreed'), and the Agreenskills fellowship programme (grant agreement n° 3215), which has received funding from the EU's Seventh Framework Programme under the agreement N° FP7-609398. The funders had no role in study design, data collection and analysis, decision to publish, or preparation of the manuscript.

**Competing interests:** The authors have declared that no competing interests exist.

**Abbreviations:** G-I, group versus individual; GWAS, genome-wide association study; NIL, near-isogenic line; SNP, single nucleotide polymorphism.

leveraging rates of phenotypic change to unprecedented levels [1,2]. At the same time, large-scale, high-throughput phenotyping platforms have become available and allow for the simultaneous quantification of multiple traits in ever larger greenhouse and field settings [3]. Yet, some studies suggest that current rates of yield increase are insufficient to meet growing demands from increasing human and animal populations under a changing climate [4,5].

Historically, the highest rates of yield increase were achieved in the middle of the 20th century, at the beginning of the "Green Revolution." Combining breeding with improved management, yield potentials of major crops, such as wheat and tropical rice, approximately doubled in only a few plant generations [6–8]. In retrospect, these gains in yield potential have been unusually large. In contrast to most classical breeding that operates through selection on polygenic variation, they were largely realized by capitalizing on single genes, notably by the introgression of discrete but pleiotropic dwarfing alleles with major effects on plant form and function [9,10]. These alleles resulted in smaller and less bushy individuals, which diverted less resources to competition and allocated more into reproduction, i.e., into grain yield. The resulting more "cooperative" genotypes could be cultivated in high-density stands, which resulted in an increased crop yield per unit land area, with little productivity increase at the individual level. In other words, a trade-off between individual fitness and group-level yield was exploited to promote yield in well-fertilized high-density stands [8,11–17]. This trade-off, however, also causes the traits that benefit the group to be selected against in mixed or segregating populations containing larger, more competitive individuals, where cooperative individuals have lower relative fitness [14,18].

Thus, a key lesson from the "Green Revolution" is that selection for competitive individuals must be avoided to achieve the highest-possible yields. One way to achieve this in breeding is to anticipate "ideotypes" with a suite of phenotypic traits that reduce competition and thereby promote cooperation. Traits desirable for the breeder are, for example, a short stature, erect leaves, and a compact root system [12,13,19]. However, a limitation of ideotype breeding is that the specific trait variation relevant for high crop yield in high-density groups often remains elusive to the human observer, especially for belowground traits. In principle, this can be avoided by shifting the focus on selection for quantitative genetic variation for cooperative traits at the group level [20–22], but this is often impractical (especially early in the breeding cycle), requiring more land and resources for phenotyping. Further, when genetic variation for the trait of interest does not exist in the breeding population, it first must be discovered in a wider crop population or in wild relatives (e.g., secondary gene pool).

Cooperative strategies in natural plant communities may be selected for under certain circumstances [e.g., spatial clustering of closely related individuals, or in cases where individuals recognized close relatives (kin recognition) and reduce competitive responses] [23,24], but when competing individuals are unrelated, the evolutionary stable strategy generally is one that maximizes individual performance (and resource use) to the detriment of the group as a whole [25–28]. However, if genetic variation for cooperation exists in nature, for whatever reason, it could potentially be leveraged in breeding [29]. The difficulty, however, is to discover relevant alleles in a diverse set of plants, in particular if they do not correspond to classical ideotypes. Here, we present a methodological framework for the discovery of such alleles. We aimed for such a framework to be as general and unbiased as possible in order to detect yield or biomass gains (or any other performance measure) that emerge from any type of cooperation, including for unknown resources and through unidentified traits. Specifically, we designed competition experiments and corresponding analytical tools that allow to rank plant genotypes on a scale ranging from "competitive" to "cooperative." We then applied these methods in a proof-of-concept experiment with a population of *Arabidopsis thaliana* genotypes and produced a genetic map of a group versus individual (G-I) performance trade-off.

Variation at one genomic region was strongly associated with variation along this trade-off, whereby the allele of minor frequency promoted cooperation. However, the identified genetic variants were not associated with individual, or monoculture-level, productivity. They only became evident when mapping the G-I performance trade-off itself. In separate experiments, we then showed that the associated genetic polymorphisms allowed predicting productivity responses of genotypes along plant—density gradients. We argue that the method we present here has a large application potential, not least because of recent advances in technology including genome-wide association studies (GWAS) and large-scale phenotyping that make these methods widely applicable.

## Results

### Quantifying a plant group-versus-individual (G-I) performance trade-off by growing genotypes in different social contexts

The core principle of our method is to measure the performance of a set of genotypes in different "social contexts" [30], allowing for the quantification and subsequent genetic mapping of variation in a G-I trade-off (Weiner [30] discusses this trade-off in detail). To achieve this, we grew individuals of a set of genotypes (1) in monoculture, i.e., we confronted them with their own social strategy; and (2) in competition with a set of "tester" genotypes that represent the range of social strategies present in the population. According to game theory [28], the most cooperative genotype of the set will perform best with similarly cooperative neighbors (i.e., as a group in monoculture) but will lose when facing selfish, highly competitive neighbors (i.e., as an individual in mixtures). Conversely, the most competitive genotype will perform worst when grown with similarly competitive neighbors but will fare well when facing a cooperative neighbor. Comparing the performance of a genotype in monoculture relative to the performance with the tester set therefore allows to quantify the genotypes' G-I performance trade-off (**Fig 1**) and thereby rank all genotypes on a continuum between cooperative and competitive. In principle, this approach is applicable to any species, but for simplicity we tested it with an association panel of 98 natural *A. thaliana* genotypes—a subset of the RegMap population [31]. Aboveground dry matter production (shoot biomass) served as the measure of performance (Materials and methods), but any other target characteristic, such as agricultural yield, could also be used. Each of the 98 focal genotypes was grown in a pot that contained another congenotypic individual (genotype monoculture). We further grew individuals of all genotypes in full factorial combination with an individual of each of ten tester genotypes (genotype mixtures; **Figs 1A and 1B** and S1). This design was replicated in two blocks.

As expected, competitive interactions among individuals were strong, with large negative effects of average tester shoot biomass (average across all pots) on shoot biomass of the focal genotypes (ANOVA $F_{1,960} = 88.23$; $P < 0.001$). To evaluate the G-I performance trade-off of genotypes, we related the monoculture shoot biomass of the target genotype to the average individual biomass of the same genotype grown in competition with tester genotypes (average individual performance in mixed stand; **Fig 1C**). Not surprisingly, across genotypes, group and individual performance were highly positively associated, with more vigorous genotypes producing more biomass both in monoculture groups and as individuals subject to competition by testers. This relationship was also nonlinear (second degree polynomial; $F_{1,95} = 8.4$, $P = 0.005$ for quadratic term), possibly because space became progressively limited as plant size increased [30,32]. However, the exact nature of this deviation from linearity is unimportant since we treated the overall relationship as heuristic. We then used the distance of each genotype's data from this empirical relationship to locate each genotype on an orthogonal axis that quantified the G-I trade-off (Materials and methods and **Fig 1C**). In other words, this

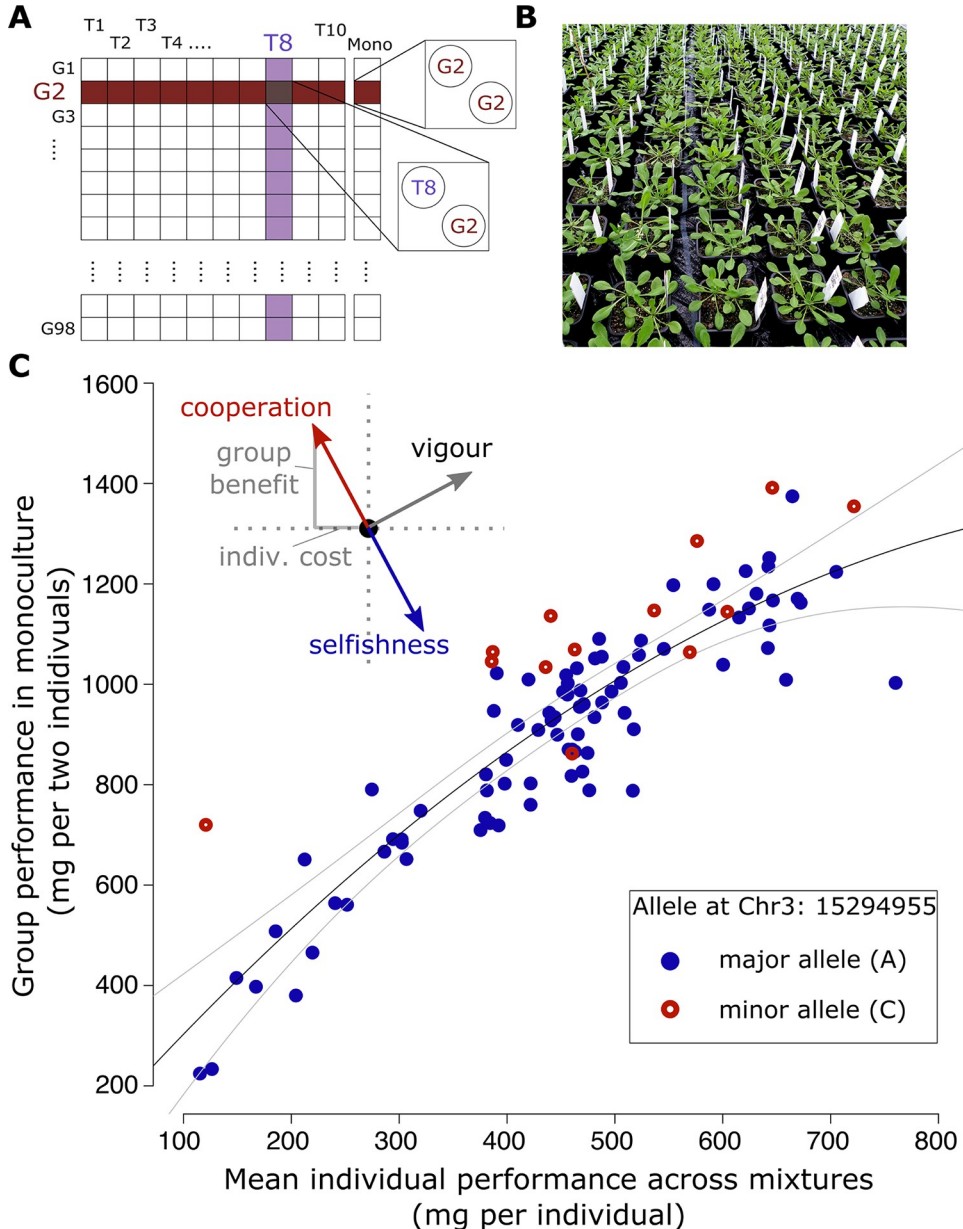

**Fig 1. A general method for the genetic dissection of the group-vs-individual performance (G-I) trade-off. (A)** Experimental design of the competition experiment. G1, G2, . . . G98: focal genotypes sampled from the RegMap panel of natural *A. thaliana* genotypes. T1, T2, . . . T10: tester genotypes representing a range in plant size to reflect a large portion of the genetic variation present within *A. thaliana*. **(B)** Experimental setup. **(C)** Relationship between a genotype's mean performance as an individual across all mixtures with tester genotypes (average individual shoot biomass) and its group performance in monoculture (average combined shoot biomass of the two individuals in a pot). Black line: second degree polynomial with 99% confidence intervals. The inset outlines three of many possible genetic effects a hypothetical allele could have on a genotype's strategy: variation along the mixture-vs-monoculture regression describes variation in vigor, and variation perpendicular to the regression describes the G-I performance trade-off. Red and blue dots show genotypes carrying different alleles at position 15'294'955 on chromosome 3 (see below), whereby red alleles confer a more cooperative strategy. Data available at https://zenodo.org/record/6983283, file competition. csv.

procedure transforms the separate values for group performance in genotype monoculture and mean individual performance in genotype mixtures into two orthogonal metrics: the position along the overall relationship reflects general genotypic vigor (e.g., increased productivity due to better adaptation to the specific growth conditions), and the position perpendicular to the general relationship reflects a G-I trade-off value that characterizes the communal properties of the focal genotype (inset **Fig 1C**). Specifically, the G-I value is positive for more cooperative genotypes, which have relatively lower individual performances in mixtures (competitive environment) but a higher performance in monoculture (cooperative environment).

## Genome-wide association mapping identifies allelic variation associated with G-I trade-off

Next, we performed genome-wide association tests for the genotypic G-I trade-off value. Genome-wide polymorphism data of our population were available through the RegMap panel, and single nucleotide polymorphism (SNP) information was available for 214,000 sites. The G-I trade-off value was significantly associated with a major effect locus on chromosome three (**Fig 2A and 2B**). The rarer allele (C-allele) was found in 18% of the RegMap population and was associated with lower individual but higher group performance, i.e., with increased cooperation (**Fig 1C**). The SNP with the strongest association resides in the center of a transposon-rich region and explained approximately 25% of the variation in the genotypic G-I trade-off values (**Fig 2C**). Direct mapping of untransformed data, i.e., of variation in either individual or monoculture group biomass alone, did not reveal any significant association with this locus (S2A and S2B Fig). This was expected, because such an analysis fails to separate general vigor from the G-I trade-off value that we used here to quantify cooperative properties. We did not find any other associations, likely because monoculture productivity is a complex polygenic trait and depends on a broad range of underlying processes. A more detailed genomic analysis based on a subset of 68 genotypes for which genome-wide resequencing data are available [33] revealed association signals across many polymorphisms in a region of approximately 150 kb around the identified RegMap SNP, all in high linkage disequilibrium (**Fig 2B**).

## Cooperative allele improves monoculture productivity at high density

To experimentally validate our finding of enhanced cooperation (i.e., of improved group performance in lines carrying the minor allele at SNP 15'294'955), we performed a stratified sampling of genotypes for plants of different sizes carrying either allele and grew these in monoculture stands of increasing planting density. Cooperative plant traits typically improve group performance especially at high planting densities [15,34,35]. Genotypes carrying the cooperation-associated allele at SNP 15'294'955 indeed exhibited superior productivity at the highest sowing density (+15% biomass, average across all genotypes; **Fig 3A and 3B**; ANOVA $F_{1,10.6} = 7.5$, $P = 0.02$), despite slightly lower individual performance across mixtures in the competition experiment (−4% biomass in mixtures; **Fig 3B**). As anticipated, group performance gains were more pronounced at higher densities where plants experienced more severe space limitations relative to those at lower densities (**Fig 3A**; ANOVA $F_{1,14.9} = 7.0$, $P = 0.019$ for allele × ground area per individual). These results demonstrate that the molecular method presented here is able to predict group-level features that cannot be deduced from individual-level properties and that these allow improving monoculture stand productivity.

## Reduced root allocation is associated with enhanced cooperation

We next explored the relationship between the allele for cooperation and phenotypic traits to identify potential mechanism underlying enhanced cooperation. We measured two traits that

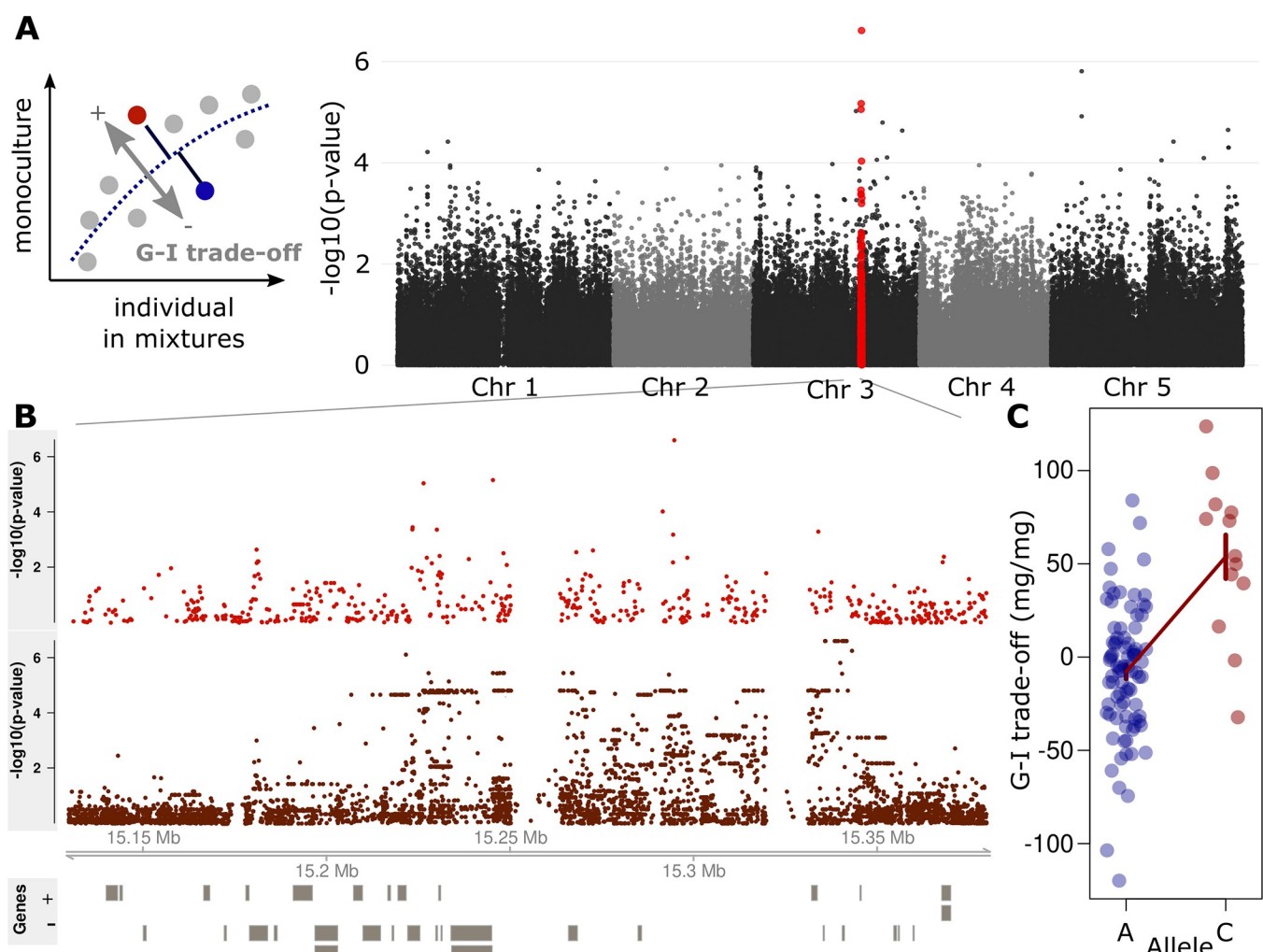

**Fig 2. Allelic variation at a major effect locus affects the group-vs.-individual performance (G-I) trade-off in *A. thaliana*. (A)** Manhattan plots of genome-wide association tests for variation in the G-I trade-off, based on the 250k SNP chip data. The genotypic G-I trade-off value is the distance from the overall trend between group and individual performance in monoculture and mixtures, respectively (inset). **(B)** Zoom in on a segment of chromosome 3, showing Manhattan plots of either an association analysis using SNP chip polymorphisms (top), or, for a subset of 68 genotypes, genome-wide resequencing polymorphisms (bottom). Models of protein-coding genes are drawn as boxes below, on either + (upper) or –(lower) strand. **(C)** Association of variation at SNP 15'294'955 and the G-I trade-off. Error bars denote means ± SEM. Data available at https://zenodo.org/record/6983283, file competition.csv.

characterize growth and resource-acquisition strategies of the genotypes in our panel. Specifically, we measured rosette diameter, because rosette lateral size affects light harvesting, and monoculture root-to-shoot ratio as an indicator of relative investment into belowground nutrient acquisition (Materials and methods). We further included two publicly available phenotypic traits into our analysis [36], namely flowering time in the field and vegetative growth rate. Genotypes carrying the allele for cooperation had significantly lower root-to-shoot ratios (ANOVA $F_{1,95}$ = 5.13, $P$ = 0.026; effect size: 19% lower root-to-shoot ratio), and these were significantly negatively associated with their G-I trade-off value (ANOVA $F_{1,95}$ = 18.4, $P$ < 0.001; **Fig 4A and 4B**). None of the other traits examined showed such an association (**Figs 4A and S3**). We confirmed that this pattern of lower root-to-shoot ratios in cooperative genotypes holds across environments by conducting an independent experiment, where we explored this effect in both monocultures and in isolated individual plants, and in a different soil type (**Figs**

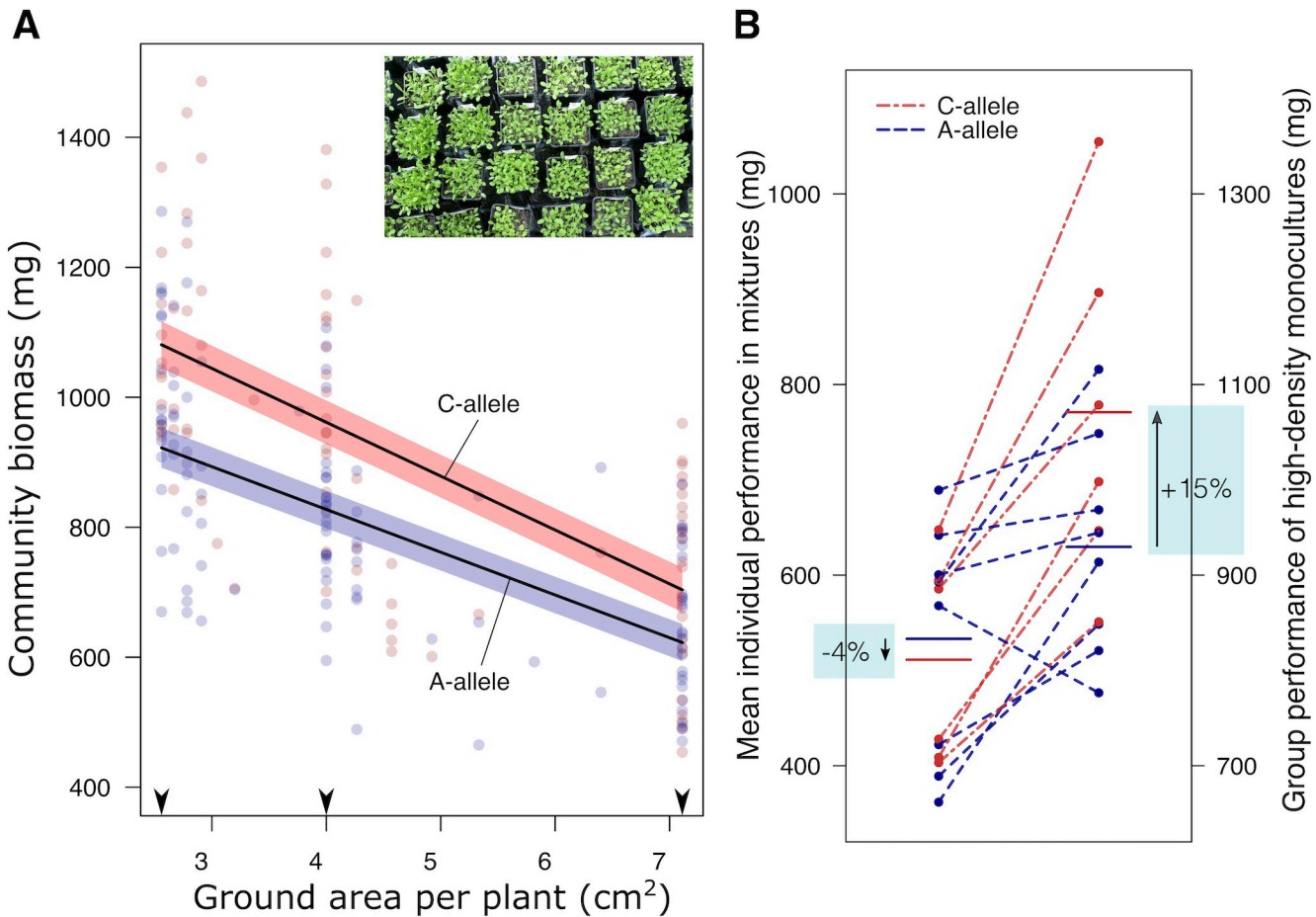

**Fig 3. Genotypes carrying the allele for cooperation exhibit superior monoculture performances at high density. (A)** Changes in monoculture biomass of genotypes carrying either the cooperation-associated allele (C-allele, red) or the alternative allele (A-allele, blue) across a realized planting density gradient. Lines show linear regression estimates ± SEM. Arrowheads show per plant areas at the sown target densities, which were not always realized due to seedling mortality. **(B)** Comparison of genotype's mean individual shoot biomass in mixtures versus monoculture biomass at densities of 25 plants per pot (2.56 cm² per plant). Horizontal lines: mean values across all genotypes carrying either allele. Red and blue: cooperation-associated allele (C-allele) and alternative allele (A-allele) at SNP Chr 3 15'294'955, respectively. Note the different scales of the left and the right y-axes. Data available at https://zenodo.org/record/6983283, file densitygrad.csv.

4C and S3). Overall, our findings indicate that relative root growth is associated with the identified genetic variation and that reduced root allocation may be part of a strategy associated with enhanced cooperation [37].

## The cooperative allele is geographically widespread and associated with pathogen resistance

Evolutionary game theory predicts that an allele that promotes cooperation will be selected against in a natural plant population, except under special circumstances [38]. We were thus surprised that the allele for cooperation that we identified is found over a wide geographic range and at remarkably high frequency (**Fig 5A**; minor allele frequency 18%). Genes often have multiple functions and we therefore expected that, in natural populations, conflicting selective forces acting on such pleiotropic genes (or on genes in tight linkage) might underpin the persistence of alleles for cooperation [39]. Examining genes in the identified genomic region, we found

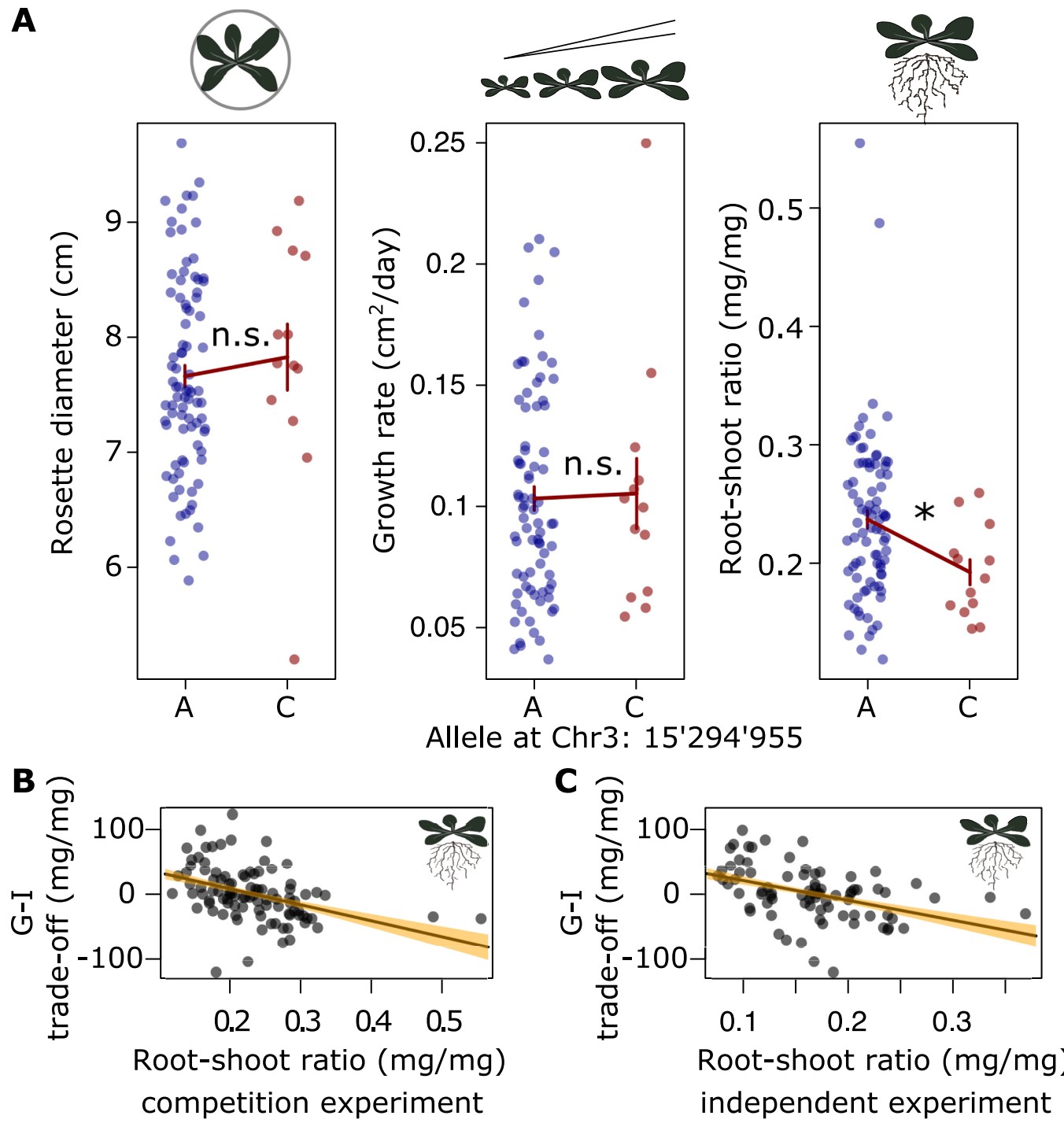

**Fig 4. Altered allocation to roots but not growth or life history is associated with increased levels of cooperation. (A)** Association of allelic variation at SNP Chr3:15'294'955 with variation in traits related to different plant strategies. **(B** and **C)** Relationship between the G-I performance trade-off and plant root-to-shoot ratio in monocultures of the competition experiment shown in Fig 1 (**B**) or monocultures of an independent experiment (**C**; see also **Materials and Methods** and **S3 Fig**). Bars and regression lines show means ± SEM; * ANOVA $P < 0.05$; n.s.: not significant. Data available at https://zenodo.org/record/6983283, files competition.csv and sand.csv.

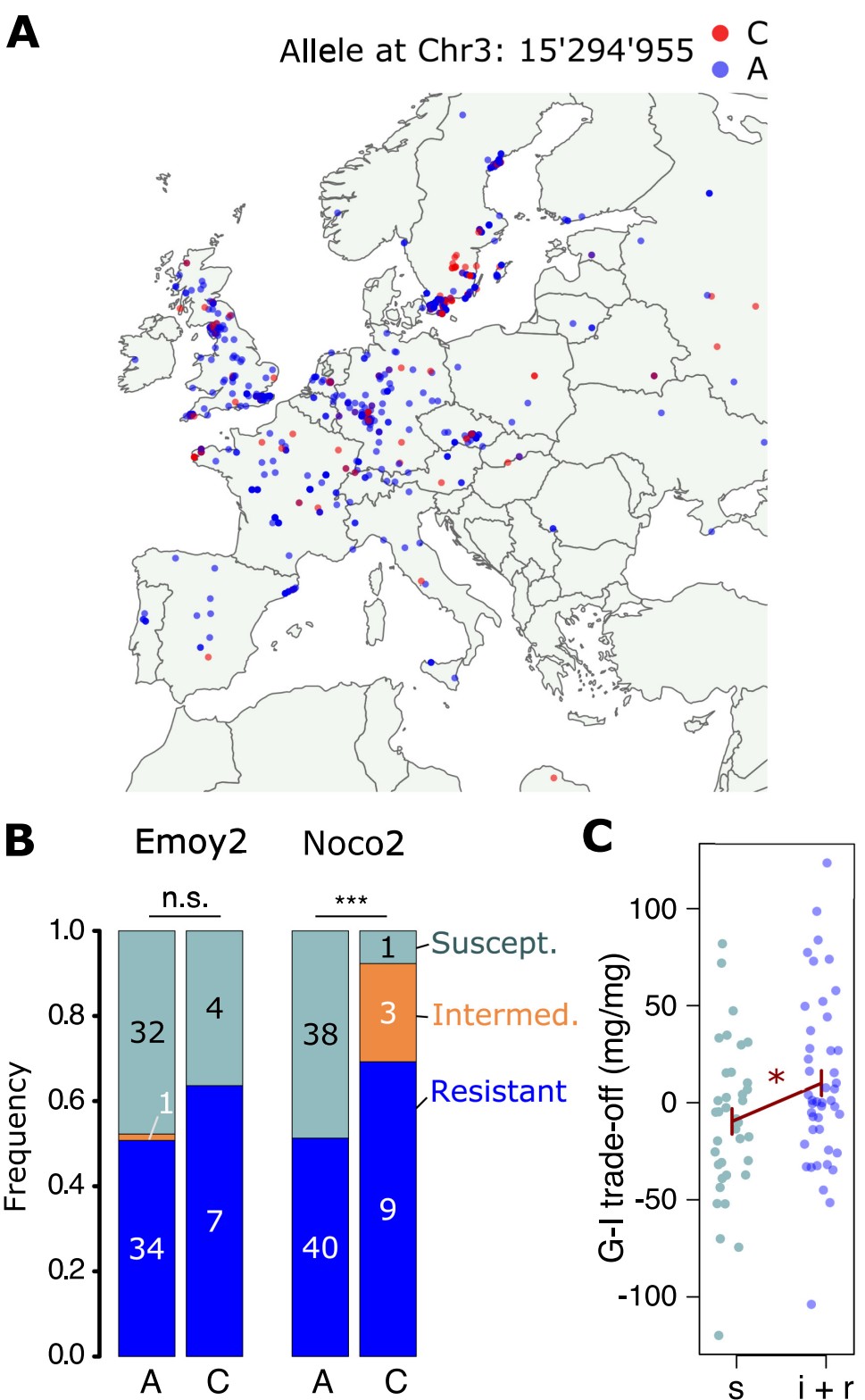

**Fig 5. The cooperation-associated allele exhibits a wide geographic distribution and is correlated with increased race-specific pathogen resistance. (A)** Occurrence of natural *A. thaliana* genotypes carrying the allele for cooperation (red 'C') or the alternative allele (blue 'A') across sampling sites in Europe. R package "maps," version 3.4.0, was used to draw the map, which uses the database from the Natural Earth data project, and all data are in the public domain

(https://cran.r-project.org/web/packages/maps/readme/README.html) **(B)** Association of Chr 3 SNP 15'294'955 with resistance or susceptibility against strains of *Hyalonperonospora arabidopsidis* (Emoy2 and Noco2), based on published data [41]. Numbers indicate genotype counts. n.s. = no significant difference. **(C)** Association of Noco2 resistance levels with the G-I trade-off. s = susceptible, i+r = intermediately and fully resistant. Data available in Supplementary File of Nemri and colleagues [41].

*AtMIN7*, a documented regulator of both growth and defense. The AtMIN7 protein targets pathogen effectors that suppress the plant immune response [40]. Importantly, variation at the *AtMIN7* gene is associated with resistance against *Hyalonperonospora arabidopsidis*, an obligate pathogen of *A. thaliana* causing downy mildew [41]. Analyzing this published data on resistance against different strains of *H. arabidopsidis* (incl. Noco2 and Emoy2*)*, we found that the cooperation-associated allele indeed was highly significantly associated with partial or full resistance against strain Noco2 (**Fig 5B**, Fisher's exact test; $P < 0.001$). The resistance level against Noco2 also explained significant amounts of variation in the G-I trade-off value of our genotypes (ANOVA $F_{2,79} = 3.57$, $P = 0.03$; **Fig 5C**). We therefore refer to this naturally occurring genetic variation as latent variation for cooperation, since contributions to pathogen resistance rather than cooperation has likely maintained the minor allele in *A. thaliana* populations.

## Lines carrying the cooperator allele have a competitive advantage in conditions of high disease pressure

Further evidence for the idea that the allele for cooperation is advantageous in environments with pathogens emerged from an additional experiment in which we quantified benefits of individual alleles (payoffs) in well-defined competitive interactions modeled according to the classic prisoner's dilemma. For the experiment, we isolated four different pairs of near-isogenic lines (NILs) from Shahdara Bayreuth recombinant inbred lines [42], i.e., pairs of nearly identical genotypes, in which allelic variation was restricted to the region around the focal SNP (see **Materials and methods** for details). It is predicted that NILs homozygous for the Shahdara (Sha) allele exhibit a more selfish and competitive, and lines homozygous for the Bayreuth (Bay) allele a more cooperative strategy. To separate effects of the Sha and Bay alleles from the general genetic background of the respective NILs, we replicated this design with four different pairs of NILs, i.e., in four different genetic backgrounds. In accordance with classic social dilemmas in evolutionary game theory (e.g., prisoner's dilemma) [28], we assumed the competitive genotypes would receive the highest payoff when growing with a cooperative neighbor (**Fig 6A**). However, such payoffs are difficult to quantify directly, and, therefore, we refer to payoffs as the reduction in aboveground biomass of plants competing relative to plants grown individually (e.g., a small reduction in biomass under competition = high payoff). We thus ran replicated experiments matching focal plants with neighbors so that all four possible combinations of Sha and Bay alleles were realized. All genotypes were also grown individually without neighbor, and the relative reduction of individual performance in competition was used to quantify the cost of growing with a specific neighbor.

Unexpectedly, the experiment was invaded by a parasite that affected genotypes and allele carriers differently and influenced competitive outcomes. Over the course of the experiment, some plants showed early leaf blotching, and, at harvest, many showed senesced and wilted leaves (**Figs 6B and S5**). This is a phenotype typical of soil-borne diseases. The symptoms were dependent on genotype and were mainly seen in NIL-background 33RV113 and individuals carrying the competitor (Sha) allele (**Fig 6C**). At the same time, we detected abundant spores of the obligate plant parasite *Olpidium brassicaea* in roots of all sampled pots ($n = 18$; **S5 Fig**). We took advantage of this to test the hypothesis that the allele increasing cooperation also

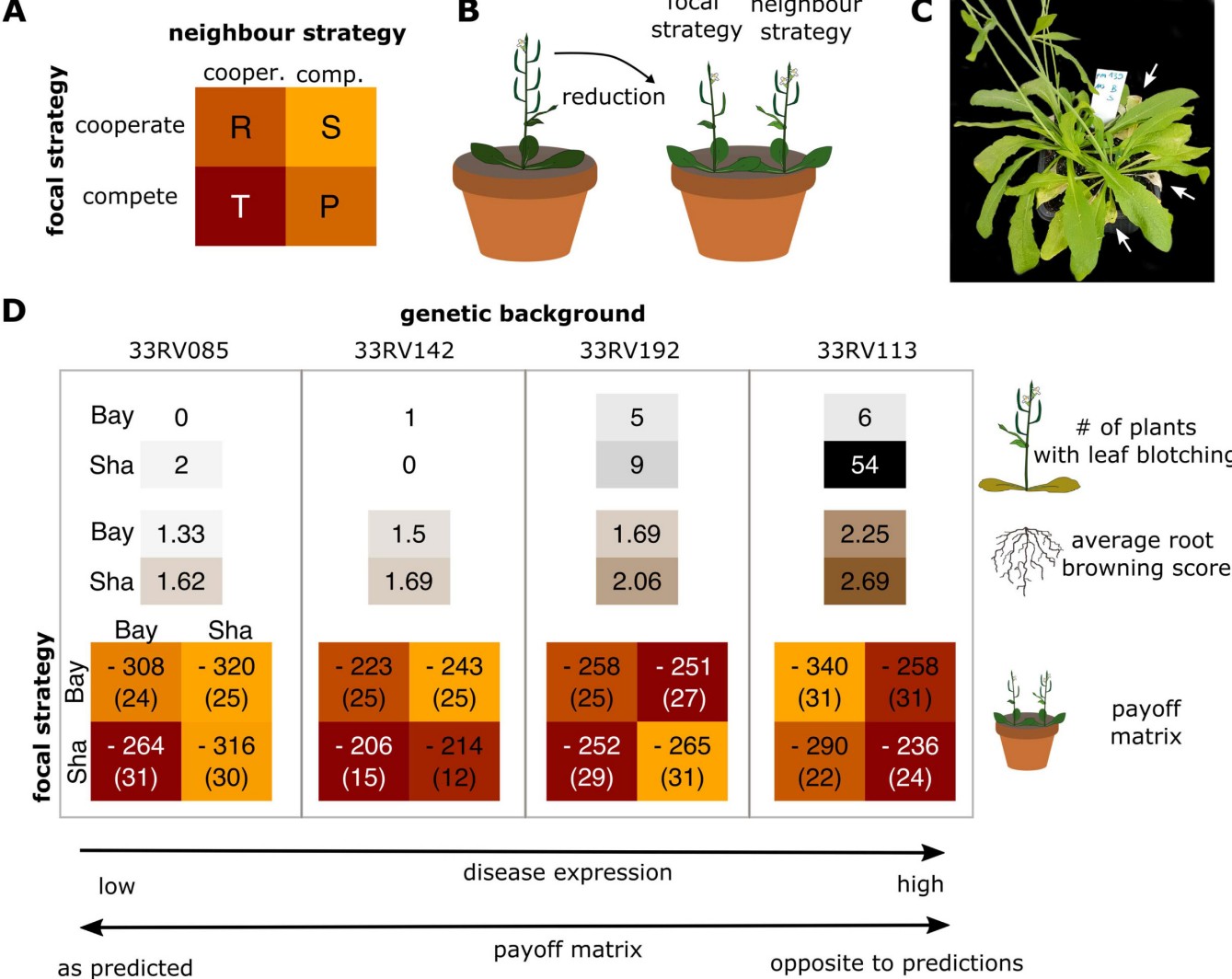

**Fig 6. A trade-off between competitive dominance and disease resistance expressed in different near-isogenic lines.** (**A**) Expected payoff matrix of interactions between plants ("cooperate" vs. "compete," as determined by carrying Bay vs. Sha alleles), with Temptation (T) > Reward (R) > Punishment (P) > Sucker's Payoff (S) (color coded). (**B**) Experimental outline: benefits refer to the cost reduction as measured by the average aboveground biomass of genotypes in competition compared to the biomass of the plant grown alone. A strong reduction refers to a high costs and therefore low benefit, a weak reduction to a high benefit. (**C**) Pot containing a pair of near-isogenic plants isolated from the RIL parent 33RV113. The plant carrying the competitive Sha allele exhibits strong signs of leaf wilting (arrows). (**D**) Summary of the competition experiment, separated by genetic background (33RV085, 33RV142, 33RV192, 33RV192), showing disease symptoms (number of plants exhibiting leaf blotching; root browning score) and payoff matrices for all genetic backgrounds and allele combinations. The values in payoff matrix are the average biomass reduction ± SEM in brackets. Data available at https://zenodo.org/record/6983283, file Payoff_longFormat.csv.

conferred pathogen resistance. As a proxy for root disease expression, we visually scored root yellowing/browning, a phenotype caused by *O. brassicae* [43]. We found that this score significantly depended on genetic background (ordinal logistic regression *p*-value < 0.001 for NIL pair) and was always lower in genotypes carrying the cooperative (Bay) allele (ordinal logistic regression *p*-value = 0.001; **Fig 6C**).

As predicted, plants carrying the allele promoting cooperation were competitively inferior when matched with a competitive neighbor, but only in the two genetic backgrounds that showed low disease expression (**Fig 6D**). In the two genetic backgrounds with high disease

expression, the competitive relationship was very different: It was most pronounced in the most susceptible genetic background, the genotype carrying the allele promoting cooperation was competitively superior.

Together, our results suggest that the cooperative allele is associated with increased resistance against several diseases, which may confer a competitive advantage under high disease pressure. This effect may be caused by antagonistic pleiotropy or by different alleles at closely linked genes. Whether or not such a trade-off between competition and disease resistance affects selection in nature will have to be tested. However, it could explain why a cooperative allele is maintained in a natural population and over such a wide geographic range.

## Discussion

Here we used specifically designed competition experiments, combined with genetic association mapping, to identify (1) plant genotypes that perform well when grown in high-density stands; and (2) potential genes and alleles that underlie these cooperative growth strategies. Interestingly, in *A. thaliana*, we detected a specific allele that is associated with increased growth in high-density stands, representing a strategy we here refer to as "cooperation." The importance of cooperation among individuals for agronomic production has long been recognized. Animal breeding has focused more explicitly on cooperation and social strategies [20,44]. Domesticated animals are often kept at relatively high densities, which can trigger aggressive behavior among individuals, resulting in high levels of stress, injuries, and increased mortality, all of which are detrimental to animal welfare and productivity. A breeding objective thus has been to obtain relatively docile, cooperative animals that limit adverse interactions when kept at high densities [45]. Conceptually similar considerations apply to plant communities. However, in plants, the traits that underlie cooperation are less easily recognized. In general, such traits may be related to resource sharing in the sense that resources remain accessible to the individuals that most efficiently convert these into biomass or yield. In other words, any excessive investment into competition in the presence of neighbors undermines what is, from an agronomic perspective, an efficient use of resources at the group level [25,26,46–50]. In plants, cooperative growth strategies thus may be characterized by a relative insensitivity to the presence of neighbors, or even a reduction in the development of competitive structures. Indeed, improved group functioning was observed in crop experiments where genotypes exhibited more vertical leaf angles [51], where leaf area [52] or branch number were reduced [53], or where belowground competition was experimentally prevented [46,50]. Cooperative responses may also be plastic and expressed predominantly in monocultures; for example, kin recognition studies have shown that plants competing with close relatives reduced root allocation relative to when they competed with strangers [54]. Common to all these experiments is the finding that restricting competitive interference between neighbors, either experimentally or by using genotypes with corresponding traits, is beneficial for group yield, especially at high planting densities. In our study, we observed a lower relative investment into root growth in cooperative genotypes, which is in line with this consideration.

Cooperation, although beneficial for group yield, is typically also associated with individual fitness costs in noncooperative environments, i.e., under strong competition [28–30]. Thus, addressing this G-I trade-off is key to identifying and using cooperative genotypes. In breeding, different strategies have been developed that achieve this goal. First, cooperation can be selected for by anticipating corresponding ideotypes possessing traits expected to foster cooperative strategies. This has been very successfully accomplished during the Green Revolution [6,12,13,55] and led to the recognition of the relevance of a G-I trade-off in crop improvement [14,18]. Second, cooperative strategies can be directly selected for at the group level [17,22,56]. For example,

this is often done in field trials with maize, whereby identical hybrid genotypes are planted and evaluated in high-density rows and subsequently in larger field plots. Such a selection at the group level also promotes traits that benefit group yield, for example, reduced tassel size, more vertical leaves, and reduced tillering [17,29,57]. Another idea is to incorporate kin selection schemes into the breeding process (e.g., in the early cycles of breeding programs where individual genotypes compete with cosegregants), which could also select for crops that exhibit cooperative traits [24,56,58]. Though it is currently unclear how widespread kin recognition is in plants [59], kin recognition studies in wild populations are another avenue that could promote the identification of cooperative traits in plants [22]. Finally, genomic predictions are increasingly adopted to predict a genotype's merit purely based on its genome, whereby prediction models are trained on yield variation of high-density stands of pure varieties. These approaches have arguably contributed to historical and contemporary yield advances and implicitly or explicitly include elements that selected to some extent for cooperative traits [22]. However, they also rely on genetic variation in the relevant traits to be present in the breeding populations. The method we present here—while motivated by the same broader topic—differs in important respects. First, it directly assesses the G-I trade-off rather than group yield. By exposing genotypes to two "social environments" that represent different ends of the cooperation–competition continuum, genotypes expressing cooperative traits can more easily be distinguished. This contrasts with other methods that either focus only on groups (e.g., row or plot-level selection) or use an environment with mixed neighborhoods (e.g., typical setting in quantitative genetic analyses of indirect genetic effects; [20,60,61]). Second, our method systematically decomposes group yield into two effects: general vigor and cooperation. This separation is critical because alleles promoting cooperation may be found at low frequency and in genotypes with low vigor. A prominent example is the recessive semidwarf allele *sd1* in rice. It was originally found in the Taiwanese Dee-Geo-Woo-Gen variety, which showed only average yields in the tropics, but was instrumental for boosting tropical rice yields [9,10,18]. Third, the fact that we use an empirical, heuristic relationship to correct for general vigor implicitly accounts for nonlinear relationships between genetics and productivity, which is typically difficult to address with other methods.

In combination with genetic mapping, our method provides an "agnostic" approach to discover latent genetic variation for cooperation. In our proof-of-concept study, we identified a genetic variant of large effect. This was surprising, since traits underlying individual growth and yield or environmental adaptations, such as drought tolerance, are often highly polygenic [62–64]. On the other hand, there is now increasing evidence that different plant–plant interactions can be affected by few genetic variants of large effects ([65,66]; this study), maybe resembling the pattern more generally found in plant–parasite interactions [67]. Regardless of the reason, the existence of such alleles is important given that many biotechnological methods, including genome editing, typically rely on the availability of traits that are influenced by a single or few genes. Therefore, we propose that biotechnological and breeding research should extend their focus beyond the performance of individuals, often grown as single plants in pots on "conveyor belt systems," towards the performance of groups and the relevant competitive interactions that drive it.

In this work, we focused—primarily for the sake of simplicity—on a model population of *A. thaliana* and on aboveground biomass (which is a target trait for some crops, such as forage grasses). This is a first case study using the proposed methodology, and a larger-scale systematic search may reveal alleles with comparable effects in crops or their wild relatives. Here, we found that the allele for cooperation among plants remained relatively frequent (18%) in natural populations, likely because it conferred resistance to diseases to the individuals carrying this allele, and thereby higher fitness under high disease pressure. This finding raises hopes that crops and wild plants may also hold hidden cooperative traits that have persisted despite their

disadvantages in competitive situations. Once identified, such latent variation in cooperation could rapidly be co-opted in marker-assisted breeding programs or through biotechnology. At a more fundamental level, the finding that large-effect genetic variants for cooperation are maintained in natural populations leads to the intriguing thought that social traits could arise as evolutionary exaptations, i.e., by co-option of an existing trait unrelated to cooperation [68].

## Materials and methods

### Plant material

The natural *A. thaliana* genotypes used (S1 Data) are a subset of the RegMap population [31] for which a comprehensive list of traits has been collected [36]. For pairs of NILs isolated from four Bay-Sha RILs, we confirmed homozygosity for a genetic polymorphism upstream of the focal SNP (primer F: 5′-TGAGAGAGAGCTGATGATGGATG-3′; primer R: 5′-CGCCTTGA TTGACACAGATTC-3′; approx. 100 bp deletion in Sha sequence at position 15'333'951) and a genetic polymorphism downstream of the focal SNP by the use of PCR markers (markers primer F: 5′-GCAAGAGGGAGCTAAAGAAACAG-3′; primer R: 5′-GCCCTTATCGCCAT GAACTG-3′; approx. 50 bp deletion in Sha sequence at position 14'914'025). These deletions in the Sha sequence were predicted by the Polymorph tool (http://polymorph.weigelworld.org). All eight genomes were also resequenced on the DNBseq plattform (BGI, Hongkong) with a minimum of three Gb sequences per genome. The reads were aligned to the *Arabidopsis* Col-0 reference sequence with the use of the bwa software (version 0.7.16a) [69]; read sorting and variant calling was performed using samtools (version 1.5) [70]. Whole-genome reconstruction was performed as described previously [65], following the approach developed by Xie and colleagues [71]. Genome reconstructions were in agreement with the published marker maps of the corresponding RILs ([42]; S5A Fig). We confirmed genetic differentiation of each of the four NIL pairs at the locus of interest. Two pairs were otherwise completely isogenic; the two other pairs showed some residual genetic differences on other chromosomes (S5B Fig).

### Experimental design

Competition experiment: Pairs of individual plants were grown in small pots (6 × 6 × 5.5 cm) in a factorial design in which the 97 genotypes of the panel were each grown together with one of ten tester genotypes (Bay-0, C24, Col-0, Cvi-0, Ler-1, Sav-0, Sf-2, Shahdara, St-0, Uk-1). These tester genotypes were chosen because they constitute the parents of different publicly available recombinant inbred line populations, i.e., a genetic resource that allows for certain genetic experiments (for example, isolating NILs from heterogeneous inbred families; see Fig 6). The tester genotypes were a subset of the panel, but tester genotypes not part of the original panel would have worked equally well. Each genotype was further grown in a monoculture of two individuals. Each genotype composition was replicated twice, in separate blocks. In the second block, not enough seeds were available for line LP-2-6; in the second block, we therefore replaced this genotype by Kn-0. We thus effectively had 98 genotypes grown with the ten tester genotypes. Overall, the experiment consisted of 2,134 pots with two plants each. Each tester line was also grown as individual plant, once per block. Pots containing single plants, and pots in which one plant died at the seedling stage, were excluded from data analyses.

Density gradient: To test for decreased self-inhibition in plants carrying the cooperation-associated allele, genotype monocultures with different individual densities (3 × 3, 4 × 4, or 5 × 5 plants on 8 × 8 cm area) were constructed. Six genotypes that carried the cooperation-associated allele (Bor-4, Est-1, Mt-0, Ra-0, Sav-0, Wa-1) but varied in average individual performances were paired with seven genotypes (An-1, Br-0, Can-0, Kondara, Nfa-10, Shahdara, St-0) that carried the alternative allele but had similar average individual performances. By

matching genotypes by size, we were able to separate the size dependence of self-inhibition from the allele effect.

## Preparation of plant material and growth conditions

Competition experiment: Seeds of all genotypes were sown directly onto soil (four parts Einheitserde ED73, Gebrüder Patzer, Germany; one-part quartz sand) in February 2016. For germination, the pots of size 6 × 6 × 5.5 cm from a given block were randomly placed into trays covered with plastic lids. The two plants per pot were established 3 to 4 cm apart by sowing 5 to 20 seeds per position and, once seeds had germinated, removing surplus seedlings. Blocks 1 and 2 were sown on February 17 and 18, 2016, respectively. Plants were grown with a photoperiod of 14 hours, providing additional light when necessary. Daytime and nighttime temperatures were maintained around 20 to 25°C and 16 to 20°C, respectively. Trays were randomly rearranged within the greenhouse every 3 to 5 days. After 5 to 5.5 weeks, pots were transferred from trays onto three tables with automated watering and randomly rearranged every week. Flowering shoots of individual plants were tied to wooden sticks when they grew taller than approx. 10 cm. All plants were harvested on April 14 (Block 1) and April 15 (Block 2), 2016, i.e., approx. 8 weeks after sowing. Each plant was cut below the rosette and individually dried at 65°C for 4 to 5 days and then stored at room temperature until weighing. Roots from a pot were washed on a metal sieve, and total root mass determined after drying at 65°C for 4 days. Flowering time was determined by checking every 2 to 3 days whether flowering bolts were present that exceeded 0.5 cm height.

Density gradient: Monocultures were sown in pots of 9 × 9 × 10 cm (inner pot diameter approximately 8 × 8 cm) at densities of either 9, 16, or 25 plants per pot. Soil and growth conditions were as described above for the competition experiment. Because we observed some seedling mortality early in the experiment, realized planting density was determined from photographs taken 27 days after sowing, i.e., at a time when competition still was very limited. Aboveground biomass was harvested, dried, and weighed 54 days after sowing.

Independent biomass allocation measurements: For an independent assessment of root-to-shoot biomass ratios in the studied natural genotypes, 80 genotypes that were used in the main competition experiment were grown for 43 days either as single plants or as monoculture (consisting of four plants per pot) and in pots of 7 × 7 × 8 cm size on a mixture of one-part ED73 and four-parts quartz sand. The measurements were performed as described above.

Competition experiments with NIL pairs: We used four NIL pairs derived from 33RV085, 33RV113, 33RV142, and 33RV192. Each pair consisted of one line carrying the Bay allele and one line carrying the Sha allele. For each pair, we grew 16 monocultures per line and 32 mixtures of the two lines, all consisting of two plants per pot. We additionally grew each line as single plants per pot, also replicated 16 times. Seeds were directly sown into 6 × 6 × 5.5 cm pots filled with a 4:1 parts ED73 soil:quartz sand. Leaf blotching and excessive senescence were noticed midway through the experiment and scored 31 and 50 days after sowing. Scoring was done blindly, i.e., without information about pot or genotype identity. Shoot and root biomass were harvested 57 days after sowing, roots washed, and all biomass dried and weighed as described above. Root browning scores were determined visually by categorizing root browning into three ordinary levels (S5 Fig). Roots were stained using a 1% cotton blue in 80% lactic acid solution and examined using a Leica microscope for a visual inspection of potential root pathogens.

## Statistical analyses

All statistical analyses were performed using the statistical software R version 3.4.1 (http://r-project.org). The block-adjusted individual performance of genotypes across mixtures were

estimated using least square means from a model including just block and genotype. Monoculture biomass per individual (i.e., total average monoculture biomass divided by two) was then fitted as function of linear and quadratic forms of individual biomass, using the R-function "lm." The G-I trade-off value was determined as orthogonal distance by determining the point in the quadratic heuristic that was closest to the respective point by nonlinear minimization using the R-function "nlm." The GWAS analyses were performed with easyGWAS (https://easygwas.ethz.ch) [72], using the EMMAX algorithm [73] and using SNPs from the 250k SNP chip (http://bergelson.uchicago.edu/) or the 1001 genomes project (http://1001genomes.org/) [33]. SNPs with a minor allele frequency below 5% were removed.

To test for the dependency of the G-I trade-off value on relative root allocation, we linearly regressed the trade-off value against root-to-shoot ratio and tested for significance of the slope term. We did not transform the axes because the relationship was sufficiently linear. Note that analysis on a log-log scale, as frequently used in allometric analyses, was not appropriate here because the G-I trade-off value was centered around zero (i.e., not systematically positive).

For the density gradient experiment, productivity was modeled in dependence of the fixed terms area per individual, allele, plus their interaction (we chose area per individual (available space) instead of density (the inverse of available space) because the relationship with density was nonlinear). The corresponding random terms were genotype, and the interaction between genotype and area per individual. Individual performance in mixture of the previous experiment was included as a covariate in the model, because it was used in the design of the experiment for stratified sampling of genotypes according to size. The realized densities deviated from sown densities because of a relatively high initial mortality. Therefore, we instead used densities determined from photographs of each pot that were made midway through the experiment. Two pots were removed from the analysis because realized densities were much higher than planted densities, probably because they accidentally had not been thinned to the intended densities.

In the NIL pair competition experiment, the payoff for a given combination of strategies was determined as shoot mass of a line grown as individual plant minus its shoot mass under competition, i.e., the payoff quantified the shot biomass reduction induced by to the presence of particular neighbors. The standard error of the payoff $\sigma_{payoff}$ was calculated by error propagation as $\sqrt{(\sigma^2_{single\ plant} + \sigma^2_{individual\ |\ neighbour})}$.

Root browning scores were analyzed using an ordinal logit regression as implemented in the polr-function of the R package MASS, with ordinal root scores as independent variable and community type (single plant or monoculture), genetic background (NIL pair), and allele (Sha or Bay) as explanatory variables. Mixtures were excluded from the analysis of root browning scores.

## Supporting information

**S1 Fig. Experimental setup of the competition experiment.** (A-C) Photos show the experiment at sowing (**A**), midway through the experiment (**B**), and at harvest day (**C**). (TIFF)

**S2 Fig. Association tests for variation in (A) average individual performance across mixtures or (B) average monoculture performance.** Data available at https://zenodo.org/record/6983283, file competition.csv. (TIFF)

**S3 Fig. Associations of SNP Chr3:15'294'955 with phenotypic variation in traits related to plant strategies.** (**A** and **B**) Shoot-to-root ratios for genotypes grown in monocultures (**A**) or as individual plants (**B**) in an independent experiment and on sand-rich soil are shown. (**C**) Published data of genotypic means in flowering time in the field [36]. Bars show mean ± SEM.

$^{**}$ = ANOVA $p < 0.01$; $^{*}$ = ANOVA $p < 0.05$; n.s., not significant. Data available at https://zenodo.org/record/6983283,filesand.csv, and at https://arapheno.1001genomes.org/phenotype/86/.
(TIFF)

**S4 Fig. Decreased self-inhibition of genotypes carrying the cooperation-associated allele along a density gradient.** Lines represent reaction norms or genotypes carrying different alleles at Chr 3 SNP 15'294'955; red lines: cooperator allele carriers; blue lines: competitor allele carriers. Data available at https://zenodo.org/record/6983283, file densitygrad.csv.
(TIFF)

**S5 Fig. A trade-off between competitive dominance and disease resistance. (A)** Genome reconstructions of two near-isogenic lines (NILs) from the 33RV085 parental RIL. Viterbi paths of each NIL are shown in red (Bay-allele at Chr3:15'294'955) and blue (Sha-allele), together with published marker data of the parent (dashed line). **(B)** Genetic differences identified in whole-genome comparisons between the pairs of near-isogenic lines. All NIL pairs differ from each other at Chr3:15'294'955 (light blue: "locus of interest"). **(C)** Earliest sign of shoot-level disease expression in line 33RV113 carrying the Sha-allele. **(D)** Microscopic image of dried root samples containing resting spores of the obligate parasite *Olpidium brassicae* (arrows). Scale bar = 50 μm. **(E)** Visual scoring scheme of root browning for washed and dried root samples from the competition experiment. Scoring was performed fully blinded to the sample ID or genotype.
(TIFF)

**S1 Data. List of *A. thaliana* genotypes used in the study, their estimated productivities across mixtures and monocultures, and measured trait values.**
(CSV)

## Acknowledgments

We thank Bernhard Schmid (UZH) and Andrea Patocchi (Agroscope) for support and helpful discussions, Matthias Lutz (Agroscope) for help in identifying *Olpidium brassicae* in root samples, Cyrille Violle (CEFE) for helpful comments on the manuscript, Matthias Philipp, Daniel Trujillo, and Mariela Soto Araya for help with sowing and harvesting the competition experiment, and Matthias Furler for technical support in the greenhouse.

## Author Contributions

**Conceptualization:** Samuel E. Wuest, Nuno D. Pires, Ueli Grossniklaus, Pascal A. Niklaus.

**Data curation:** Samuel E. Wuest.

**Formal analysis:** Samuel E. Wuest, Pascal A. Niklaus.

**Funding acquisition:** Samuel E. Wuest, Ueli Grossniklaus.

**Investigation:** Samuel E. Wuest, Nuno D. Pires, Shan Luo, Pascal A. Niklaus.

**Methodology:** Samuel E. Wuest, Pascal A. Niklaus.

**Resources:** Nuno D. Pires, Francois Vasseur, Julie Messier, Ueli Grossniklaus.

**Visualization:** Samuel E. Wuest, Pascal A. Niklaus.

**Writing – original draft:** Samuel E. Wuest, Pascal A. Niklaus.

**Writing – review & editing:** Samuel E. Wuest, Nuno D. Pires, Shan Luo, Francois Vasseur, Julie Messier, Ueli Grossniklaus, Pascal A. Niklaus.

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
