## [Editor Report · Decision Letter 0]

6 Apr 2022

Dear Dr Wuest, 

Thank you for submitting your manuscript entitled "Increasing plant group productivity through latent genetic variation for cooperation" for consideration as a Research Article by PLOS Biology.

Your manuscript has now been evaluated by the PLOS Biology editorial staff, as well as by an academic editor with relevant expertise, and I'm writing to let you know that we would like to send your submission out for external peer review.

IMPORTANT: Your paper was submitted as a regular Research Article, but we think that it would be better considered as a methods paper. No re-formatting is required, but please select "Methods and Resources" as the article type when you upload your additional metadata (see next paragraph).

Once your full submission is complete, your paper will undergo a series of checks in preparation for peer review. Once your manuscript has passed the checks it will be sent out for review. To provide the metadata for your submission, please Login to Editorial Manager (https://www.editorialmanager.com/pbiology) within two working days, i.e. by Apr 08 2022 11:59PM.

If your manuscript has been previously reviewed at another journal, PLOS Biology is willing to work with those reviews in order to avoid re-starting the process. Submission of the previous reviews is entirely optional and our ability to use them effectively will depend on the willingness of the previous journal to confirm the content of the reports and share the reviewer identities. Please note that we reserve the right to invite additional reviewers if we consider that additional/independent reviewers are needed, although we aim to avoid this as far as possible. In our experience, working with previous reviews does save time. 

If you would like to send previous reviewer reports to us, please email me at rroberts@plos.org to let me know, including the name of the previous journal and the manuscript ID the study was given, as well as attaching a point-by-point response to reviewers that details how you have or plan to address the reviewers' concerns. 

Given the disruptions resulting from the ongoing COVID-19 pandemic, please expect some delays in the editorial process. We apologise in advance for any inconvenience caused and will do our best to minimize impact as far as possible.

Kind regards,

Roli Roberts

Roland Roberts

Senior Editor

PLOS Biology

rroberts@plos.org

---

## [Decision Letter · Decision Letter 1]

23 Jun 2022

Dear Dr Wuest,

Thank you for your patience while your manuscript "Increasing plant group productivity through latent genetic variation for cooperation" went through peer-review at PLOS Biology. Your manuscript has now been evaluated by the PLOS Biology editors, an Academic Editor with relevant expertise, and by three independent reviewers.

You’ll see that all three reviewers are very positive about your study; however, they each raise a number of concerns that will need to be addressed. Reviewer #1 wants you to tighten up your definition of “cooperation,” and identifies a couple of issues with the experimental design. She also recommends that you remove the resistance aspect to another paper, but we're happy to leave that decision up to you. Reviewer #2 only has minor textual requests, while reviewer #3 asks how feasible the approach would be in industry-scale breeding, wants you to speculate about non-linearity as an explanation for some results, and has a couple of other minor points.

In light of the reviews, which you will find at the end of this email, we are pleased to offer you the opportunity to address the comments from the reviewers in a revision that we anticipate should not take you very long. We will then assess your revised manuscript and your response to the reviewers' comments with our Academic Editor aiming to avoid further rounds of peer-review, although might need to consult with the reviewers, depending on the nature of the revisions.

**IMPORTANT - SUBMITTING YOUR REVISION**

*Resubmission Checklist*

*Published Peer Review*

*PLOS Data Policy*

*Blot and Gel Data Policy*

Sincerely,

Roli Roberts

Roland Roberts, PhD

Senior Editor

PLOS Biology

rroberts@plos.org

REVIEWERS' COMMENTS:

Reviewer #1:

[identifies herself as Susan A Dudley]

Wuest et al 2022 Plos Biology

This exciting manuscript provides a mechanism for screening multiple genotypes for cooperation, does so using a set of mapping lines, and finds an association between a putative cooperative allele and lowered root allocation and increased disease resistance. The question is excellent and important. 

This manuscript justifies and illustrates an approach to finding 'cooperative' alleles for crop breeding, in which 'cooperative' alleles are defined as alleles that increase stand fitness at the cost of individual fitness. They then screened for 'cooperative' alleles by growing plants in same accession vs. different accession pairs, and looked for the accessions that fit that fitness pattern. They were able to map the alleles and demonstrate that they are associated with root allocation and disease resistance. There is somewhat of a mismatch in the main experiment, though, between the goals of the researchers and what their methodology ends up measuring. I am seeing one experimental weakness and some conceptual issues.

Their definition of cooperation from game theory (line 38) meets the definition of altruism in evolutionary theory. 'Cooperative' alleles are defined as alleles that increase stand fitness at the cost of individual fitness. This definition is actually a definition of altruism, or costly help. In reference 22, I used altruism to refer to costly help, and cooperation to refer to within-species help that benefits the helper. Cooperation can be defined more broadly as any kind of within species helping, costly or not. I strongly argue that authors should reference and discuss their definition of cooperation, given the confusion in the literature.

I did not argue in my review [22] that "plant cooperation among unrelated individuals rarely evolves in natural systems because plants often grow in communities of unrelated individuals, a setting under which cooperative phenotypes have low fitness" (lines 96-98). Plants also often grow in groups of related individuals because most dispersal is local - so-called viscous populations (Hamilton). Facilitation, helping between species at the same trophic level, is frequently found in nature. 

Moreover, the authors may have wanted to find altruism towards unrelated individuals, but their experimental design is the same one used in kin recognition studies. Plants were grown in same- and different-accession pairs in the experimental design. The goal was to create a monoculture of equally cooperative or competitive phenotypes, but there is a considerable body of evidence on plant identity recognition that needs to be considered. Many researchers have measured the fitness consequences of growing with relatives (File et al. 2012), though to my knowledge not at the fine scale seen in the manuscript. This manuscript should reference (Milla et al. 2009) who measured the fitness consequences of growth in pairs of kin and strangers in Arabidopsis. 

The results in this manuscript are not necessarily a consequence of kin discrimination. Altruism can evolve in viscous populations and as an outcome of multilevel selection (Weinig et al. 2007). Direct benefit helping (=cooperation as defined in [22]), which is 'within-species facilitation' is understudied but is known to occur. The methods in the present manuscript would potentially find these types of helping. 

The authors used a factorial design to create the accession pairs, with all possible combinations of the 97 panel accessions with 10 tester accessions and itself. The analysis is not an anova; instead the authors use performance in the same accession pair as the dependent and average performance with the tester accessions as a predictor. However, the consequence is that performance in mixed stands is better replicated (20 pots per accession) than performance in monoculture (2 pots per accession) so that the standard error is higher for estimates of performance in monoculture. I'd class this as a weakness rather than a fatal flaw since it doesn't violate the assumptions of regression. Plant growth is variable, and more replication of the monoculture would be better in future studies. 

Detailed comments follow:

1. I like the introduction with the clear discussion of ideotype breeding. It motivates the study well. 

2. Line 130-134. Reference for cooperation in game theory?

3. Line 137, figure 1. What is the justification - theory and references - behind the estimates of the G-I tradeoff? This important approach needs more discussion. 

4. In figure 1, the axis indicates group performance in monoculture, while line 572 suggests the model was fit using "Monoculture biomass per individual (i.e. total average monoculture biomass divided by two). I note that in Figure 1, average individual performance in mixtures is mostly slightly less than half that of a pair of same-accession plants. Cooperative accessions are evaluated as above average rather than as having higher performance in monoculture than mixture. That one point with low performance in mixture and over 3X higher performance in monoculture may be very influential. Here the value of more replication in monocultures is clear. 

5. Use of synonymous terms - accession and genotype. Use one consistently.

6. In reading the experimental design, I found that it was unclear by how it is written that each panel genotype is paired with each tester genotype, and had to do some math to confirm that it is a factorial design. 

7. The figures are not colourblind friendly or printer friendly, since often they only indicated point identity by red and blue. Ideally they should use shape and colour to distinguish groups, with colourblind friendly colours

8. Line 230. Why was area per individual used rather than plants per square meter, when the legend indicates performance across a realized planting density gradient? Plants per square meter is more typical in ecology and can be readily compared to other studies. The increased performance with density indicates maximum yield per pot was not realized, which is good.

9. Line 261+. "We confirmed that this pattern of lower root-to-shoot ratio in cooperative genotypes holds across environments by conducting an independent experiment, where we explored this effect in both monocultures and in isolated individual plants, and in a different soil type (Fig. 4C, and Supplementary Fig. S3). Overall, our findings indicate that reduced relative root growth is part of a genetically fixed strategy associated with enhanced cooperation." Root allocation tends to be plastic in Arabidopsis to nutrients. The lateral root number increases with strange exudates compared to kin and self exudates (Biedrzycki et al. 2010) .

10. Line 295, line 461. This conclusion that any variation in cooperation must be latent seems very weak to me. It ignores considerable theory and some evidence that cooperation and altruism can be favored in plants. 

11. The resistance evidence is post-hoc. It is intriguing but not a planned study. It is also difficult to follow. Some of that is presentation. Fig 6 does not clearly indicate the reward structure is for the focal plant (rows) as determined by the strategy of both the focal plant (rows) and the neighbour plant (columns). The Wikipedia explanation of the prisoners dilemma payoffs is more clear than what is presented here. The text vs. background contrast is very low for the darkest background and I cannot read the numbers. Some of that is that is that the design is complex and needs more attention from the reader. I recommend it be published as a separate paper. 

12. Lines 511 - 518.The authors should be clear that the stands are single accession.

13. The analysis section does not include root allocation. I do promote the methods that use an additive formulation (Coleman et al. 1994; Gedroc et al. 1996; McConnaughay and Coleman 1999).

14. Reference 51 is the same as 21

Biedrzycki, M., Jilany, T., Dudley, S. and Bais, H. 2010. Root exudates mediate kin recognition in plants. Communicative and Integrative Biology, 3: 1-8.

Coleman, J.S., McConnaughay, K.D.M. and Ackerly, D.D. 1994. Interpreting phenotypic variation in plants. Trends in Ecology and Evolution, 9: 187-191.

File, A.L., Murphy, G.P. and Dudley, S.A. 2012. Fitness consequences of plants growing with siblings: reconciling kin selection, niche partitioning and competitive ability. Proceedings of the Royal Society B-Biological Sciences, 279: 209-218. http://www.doi.org/10.1098/rspb.2011.1995.

Gedroc, J.J., McConnaughay, K.D.M. and Coleman, J.S. 1996. Plasticity in root/shoot partitioning: optimal, ontogenetic, or both? Functional Ecology, 10: 44-50.

McConnaughay, K.D.M. and Coleman, J.S. 1999. Biomass allocation in plants: ontogeny or optimality? A test along three resource gradients. Ecology, 80: 2581-2583.

Milla, R., Forero, D.M., Escudero, A. and Iriondo, J.M. 2009. Growing with siblings: a common ground for cooperation or for fiercer competition among plants? Proceedings of the Royal Society of London, Series B: Biological Sciences, 276: 2531-2540.

Weinig, C., Johnston, J.A., Willis, C.G. and Maloof, J.N. 2007. Antagonistic multilevel selection on size and architecture in variable density settings. Evolution, 61: 58-67. http://www.doi.org/10.1111/j.1558-5646.2007.00005.x.

Reviewer #2:

[identifies himself as Stephen Bonser]

World food production relies on our ability to increase crop yield. Identifying genes underlying increased crop yield is an extraordinarily important research area. The authors present the results of an interesting study demonstrating a method to identify genotypes within populations that are more productive (cooperative) when gown together in high density stands. I believe this is an important study that could have wide ranging impacts on crop production. I have some comments on the current manuscript, most of which are relatively minor.

1) The main thing that I would have recommended be done differently is to present results for reproductive output or fecundity. Most crops are seeds or fruits, and reproductive output may have been a better measure than shoot biomass of performance under competition. It is likely that reproductive output is correlated with shoot mass, and the results would probably be quite similar. However, competition can induce plants to increase their allocation to reproduction at the cost of reduced future vegetative growth. The authors build a compelling case that the alleles associated with cooperation in Arabidopsis are also associated with pathogen resistance, and this can explain the maintenance of cooperation alleles in the population. If competition induced more efficient reproduction, and this was an important adaptive response in plants, then it may be common for alleles associated high reproductive yield under competition (since these alleles wouldn't have been strongly selected against as many cooperation alleles would have been). This might be worth mentioning in the manuscript.

2) Lines 85-86 - We don't yet have a great understanding of the traits associated with competitive ability in plants. These traits may be associated with poor competitive ability, but they may not be.

3) Figure 2b - this relationship seems to be driven by the two points with the highest root-shoot ratio. Consider transforming, or demonstrating that the relationship is significant without these two points. 

4) Line 326 - I may have missed this in the methods, but how was the reduction in competition measured? 

5) Line 382 - increasing density tends to decrease growth. This allele tends to allow plants have high growth relative to the genotypes with the more competitive allele.

6) Line 498 - provide details of pot size. 

Reviewer #3:

Wuest et al. present a set of elegant experiments to map loci underlying individual-group tradeoffs in Arabidopsis, and propose these approaches as a novel framework to identify loci for breeding. I really enjoyed reading this paper -- clearly thought-out experiments, good presentation, and well written. I also like the authors' overall argument about the importance of loci underlying cooperation, and understand the choice of Arabidopsis in terms of feasibility. Overall I'm very positive about this work! 

I do have a few of clarifications and questions. Likely some are from my own misunderstanding or misreading, so I apologize ahead of time for things I may have missed.

First, I am not sure I agree with the authors that their general framework is feasible for most crops. For a breeder evaluating progeny from some cross, doing this full set of experiments would be very time consuming and costly and is not easy to set up in a field scenario. Greenhouse work costs more and less scaleable than field trials. I really have trouble seeing how this experimental approach could approach industry-scale breeding.

Second, I think I could use some more clarity on the value of this approach over just selecting for group yield/fitness alone, or looking for loci by mapping fitness in a bunch of monogenotypic trials (which is e.g. what is done in maize as the authors cite in the discussion). I was surprised to see their results that suggest the chr3 locus is not found in GWAS for overall group fitness, and would like some more explanation on what they think is going on. It seems to me that, under a simple additive model, given a sufficiently powered data set you should see the effect of this locus even if it's sometimes found in low-vigor genotypes. So the argument that "cooperator" alleles can't be found in low-vigour genotypes doesn't seem to hold for me -- you can find "tall" alleles even though they are sometimes found in short individuals, for example. Instead, perhaps it is something along the lines of the nonlinearity between individual/group fitness or genotype/phenotype the authors bring up in the discussion (line 440-442) what's driving this disconnect between the overall group fitness and ability to map this locus? If so, is some kind of nonlinearity to be expected more commonly for "cooperator" loci than for other traits? Some additional explanation/guidance/speculation here would be helpful, since at least my initial reaction was that mapping/selecting for group yield is already in fact what lots of breeding is doing, making me wonder why this approach was necessary. The authors empirically show that you get a different result, but it would be helpful I think to dig in a bit more as to why and how often we might expect this to be the case.

Finally, is it really fair to call the chr3 locus a "cooperator" allele given the disease results where it leads to increased competitive fitness too (lines 349-352)? Is this a cooperation locus or just a locus that affects disease resistance and has a pleiotropic effect decreasing root/shoot ratio (or does it just affect root size alone)?

Minor Points:

The extended LD around the locus on chr3 sure looks like this could be a structural variant. I recommend the authors look up the region in assembly comparisons of thaliana lines to see (I looked in https://academic.oup.com/mbe/article/38/4/1498/6008718#235101396 but doesn't look to me like there's an inversion of sufficient size there, though I didn't check their insertion/deletion results). 

Does root-shoot ratio by itself predict cooperation or group fitness?

---

## [Decision Letter · Decision Letter 2]

5 Sep 2022

Dear Dr Wuest,

Thank you for your patience while we considered your revised manuscript "Increasing plant group productivity through latent genetic variation for cooperation" for publication as a Methods and Resources at PLOS Biology. This revised version of your manuscript has been evaluated by the PLOS Biology editors, the Academic Editor, and one of the original reviewers.

Based on the review, we are likely to accept this manuscript for publication, provided you satisfactorily address the remaining points raised by the reviewer. Please also make sure to address the following data and other policy-related requests.

IMPORTANT:

a) Please attend to the remaining requests from reviewer #1.

b) Please provide a blurb, according to the instructions in the submission form.

c) Please address my Data Policy requests below; specifically, we need you to supply the numerical values underlying Figs 1C, 2ABC, 3AB, 4ABC, 5ABC, 6D, S2AB, S3ABC, S4, S5AB, either as a supplementary data file or as a permanent DOI’d deposition like Zenodo, Dryad, Figshare, etc.

d) Please cite the location of the data clearly in all relevant main and supplementary Figure legends, e.g. “The data underlying this Figure can be found in https://zenodo.org/record/XXXXX”

We expect to receive your revised manuscript within two weeks. 

*Published Peer Review History*

*Press*

Sincerely,

Roli Roberts

Roland Roberts, PhD

Senior Editor,

rroberts@plos.org,

PLOS Biology

DATA POLICY:

Regardless of the method selected, please ensure that you provide the individual numerical values that underlie the summary data displayed in the following figure panels as they are essential for readers to assess your analysis and to reproduce it: Figs 1C, 2ABC, 3AB, 4ABC, 5ABC, 6D, S2AB, S3ABC, S4, S5AB. NOTE: the numerical data provided should include all replicates AND the way in which the plotted mean and errors were derived (it should not present only the mean/average values).

SPECIES INDICATED IN THE ABSTRACT? 

- Please note that per journal policy, the model system/species studied should be clearly stated in the abstract of your manuscript. 

We require the original, uncropped and minimally adjusted images supporting all blot and gel results reported in an article's figures or Supporting Information files. We will require these files before a manuscript can be accepted so please prepare and upload them now. Please carefully read our guidelines for how to prepare and upload this data: https://journals.plos.org/plosbiology/s/figures#loc-blot-and-gel-reporting-requirements

DATA NOT SHOWN?

REVIEWER'S COMMENTS:

Reviewer #1: 

[identifies herself as Susan A. Dudley]

This is an important and interesting paper. It reads very well. I strongly suggest the Figure 1 be better documented - see comments below. I also have a few detailed comments that I think would improve the manuscript. The introduction and methods could better integrate the finding from multiple researchers that plants demonstrate plasticity to the relatedness of neighbours (kin recognition). The authors are screening for cooperative alleles expressed in monoculture. Because crops are grown in monoculture, this is entirely appropriate. However, they should acknowledge that there is a potential for a kin recognition response affecting performance between monocultures and mixed pots. Though they don't analyze the monoculture/mixed pot curve, the markedly higher fitness for accessions grown in monoculture vs. mixed pots is predicted by kin selection theory, and contributes to kin recognition research, and so itself interesting. That they found a tractable single locus cooperation allele as well demonstrates the strength of their approach. While looking for kin recognition is arguably irrelevant for crops grown in monoculture, kin recognition should not be ignored for this study.

1. Line 93 -97 . Altruism (costly help) should only evolve in related groups, but cooperation broad sense evolves because it will benefit the cooperating individual. See (Lehmann and Keller 2006) (Dudley 2015). Be more exact in the use of terminology. 

2. Line 98. Population structuring is common in plants as most dispersal is local even with mechanism for long distance dispersal. Plants commonly self and/or clone, resulting in high relatedness. Therefore, conditions that select for altruism may happen frequently, depending on the species ecology. 

3. Line 129-137 makes arguments for the experimental procedure entirely on the basis of functional traits and game theory, ignoring kin selection theory and the potential for plasticity to the relatedness of neighbours. Because same accession is the proxy for same functional traits, kin recognition and constitutive cooperation are confounded. Because the goal is to identify cooperation in monoculture, this is not a problem. However, not discussing these issues will result in readers aware of the work to question the methodology.

4. The legends for figure 1 does not make it clear whether the authors are plotting mass of average single plant in monoculture or average pair of plants in monoculture vs average of a single plant in mixture. The legend needs to explain what the 'group' is. Lines 580-582 indicates that the analysis used the average individual single plant in monoculture. If the authors are plotting the average individual single plant in Fig 1, then monoculture plants are substantially larger than mixed culture plants. I would like to know if they found any size differences between mixed and monoculture plants. 

5. Line 425-426. I actually disagree with using kin selection in crop breeding itself, since crops are grown in single cultivar groups, and don't experience competition with non-relatives of the same species, so there is little opportunity for kin selection or advantage to kin recognition. However, studies of kin recognition in wild plants are another way of picking up potential cooperative traits, e.g. (Dudley and File 2007) also identifies root allocation as a competitive/cooperative trait. 

Dudley, S.A. and File, A.L. 2007. Kin recognition in an annual plant. Biology Letters, 3: 435-438. http://www.doi.org/10.1098/rsbl.2007.0232

Dudley, S.A. 2015. Plant cooperation. Aob Plants, 7. http://www.doi.org/10.1093/aobpla/plv113

Lehmann, L. and Keller, L. 2006. The evolution of cooperation and altruism - a general framework and a classification of models. Journal of Evolutionary Biology, 19: 1365-1376. http://www.doi.org/10.1111/j.1420-9101.2006.01119.x

---

## [Editor Report · Decision Letter 3]

20 Sep 2022

Dear Dr Wuest,

Thank you for the submission of your revised Methods and Resources "Increasing plant group productivity through latent genetic variation for cooperation" for publication in PLOS Biology. On behalf of my colleagues and the Academic Editor, Pamela Ronald, I'm pleased to say that we can in principle accept your manuscript for publication, provided you address any remaining formatting and reporting issues. These will be detailed in an email you should receive within 2-3 business days from our colleagues in the journal operations team; no action is required from you until then. Please note that we will not be able to formally accept your manuscript and schedule it for publication until you have completed any requested changes.

Sincerely, 

Roli Roberts

Senior Editor

PLOS Biology

rroberts@plos.org